# GenImage: A Million-Scale Benchmark for Detecting AI-Generated Image

**Mingjian Zhu, Hanting Chen, Qiangyu Yan, Xudong Huang,**
**Guanyu Lin, Wei Li, Zhijun Tu, Hailin Hu, Jie Hu, Yunhe Wang**[*]
Huawei Noah's Ark Lab
{zhumingjian, yunhe.wang}@huawei.com

## Abstract

The extraordinary ability of generative models to generate photographic images has intensified concerns about the spread of disinformation, thereby leading to the demand for detectors capable of distinguishing between AI-generated fake images and real images. However, the lack of large datasets containing images from the most advanced image generators poses an obstacle to the development of such detectors. In this paper, we introduce the GenImage dataset, which has the following advantages: 1) Plenty of Images, including over one million pairs of AI-generated fake images and collected real images. 2) Rich Image Content, encompassing a broad range of image classes. 3) State-of-the-art Generators, synthesizing images with advanced diffusion models and GANs. The aforementioned advantages allow the detectors trained on GenImage to undergo a thorough evaluation and demonstrate strong applicability to diverse images. We conduct a comprehensive analysis of the dataset and propose two tasks for evaluating the detection method in resembling real-world scenarios. The cross-generator image classification task measures the performance of a detector trained on one generator when tested on the others. The degraded image classification task assesses the capability of the detectors in handling degraded images such as low-resolution, blurred, and compressed images. With the GenImage dataset, researchers can effectively expedite the development and evaluation of superior AI-generated image detectors in comparison to prevailing methodologies.

## 1   Introduction

The advancement of generative models has yielded remarkable progress in synthesizing photorealistic images, drastically reducing the requisite expertise and effort to generate fake images. This unprecedented accessibility has evoked concerns regarding the ubiquitous dissemination of disinformation. Fake images are particularly convincing because of visual comprehensibility. Therefore they enable a negative impact on social areas such as politics and economics by manipulating public opinion. For example, AI-generated photos of the Pentagon on fire are widely shared on Twitter [22]. This image fools several major news outlets and causes a significant drop in the US stock market. Current state-of-the-art (SOTA) generative models have exhibited the ability to generate images that pose a significant challenge to human perception and discrimination. Humans can only achieve an accuracy rate of 61.3% when discriminating between real images and AI-generated fake images [19]. To this end, it is of utmost urgency to prioritize the development of an effective detector capable of accurately identifying fake images generated by these advanced generative models.

To aid in the development of detectors, early fake image detection dataset [30, 3] primarily concentrate on face forgery, utilizing generative models to manipulate the human face, thereby replacing a person's

---

[*]Corresponding Author

37th Conference on Neural Information Processing Systems (NeurIPS 2023) Track on Datasets and Benchmarks.

Table 1: An overview of fake image detection datasets.

| Dataset | Image Content | Generator Category | | Public Avalibility | Real Images | Fake Images |
| --- | --- | --- | --- | --- | --- | --- |
| | | GAN | Diffusion | | | |
| UADFV [36] | Face | ✓ | ✗ | ✗ | 241 | 252 |
| FakeSpotter [30] | Face | ✓ | ✗ | ✗ | 6,000 | 5,000 |
| DFFD [3] | Face | ✓ | ✗ | ✓ | 58,703 | 240,336 |
| APFDD [7] | Face | ✓ | ✗ | ✗ | 5,000 | 5,000 |
| ForgeryNet [10] | Face | ✓ | ✗ | ✓ | 1,438,201 | 1,457,861 |
| DeepArt [32] | Art | ✗ | ✓ | ✓ | 64,479 | 73,411 |
| CNNSpot [31] | General | ✓ | ✗ | ✓ | 362,000 | 362,000 |
| IEEE VIP Cup [29] | General | ✓ | ✓ | ✗ | 7,000 | 7,000 |
| DE-FAKE [27] | General | ✗ | ✓ | ✗ | 20,000 | 60,000 |
| CiFAKE [1] | General | ✗ | ✓ | ✓ | 60,000 | 60,000 |
| **GenImage** | **General** | ✓ | ✓ | ✓ | **1,331,167** | **1,350,000** |

identity or modifying the facial attributes. UADFV [36] presents a small-scale face forgery dataset, comprising solely 241 real images and 252 fake images. A larger dataset, namely ForgeryNet [10], introduces a more comprehensive dataset encompassing over one million instances of face forgery images. Nevertheless, the applicability of these datasets is limited due to their sole focus on face images, rendering them less effective for a broader range of image categories. Early datasets for general AI-generated images are built upon generative adversarial network (GAN), such as CNNSpot [31]. CNNSpot only employs ProGAN [12] for generating training set and evaluates detector performance across a variety of GAN-based testing set. These generators generate images bearing a strong resemblance to real images, although humans could often detect their synthetic nature. Besides GAN, the recent advent of alternative generators, such as diffusion models, significantly enhances the quality of generated images, rendering the task of distinguishing real from fake images increasingly challenging. Therefore, it is essential to explore the images generated by diffusion models. IEEE VIP Cup [29] and DE-FAKE [27] have employed diffusion models to generate more general images. Based on a small-scale Cifar10 dataset [15], CiFAKE [1] generates fake images only with Stable Diffusion V1.4. However, existing diffusion model datasets suffer from limited data.

In this paper, we generate an extensive collection of general images leveraging the current state-of-the-art Diffusion and GAN models, subsequently constructing our dataset, namely GenImage. Our target is to be on par with awesome million-scale datasets like ImageNet [4] for complete and comprehensive AI-generated image detector training and validation. We use the 1000 class labels in ImageNet to generate 1.3 million fake images, which equals the number of real images in ImageNet. In Table 1, we compare GenImage with the other datasets. Compared with the face forgery datasets, GenImage covers a broader range of image content, such as basketball and guitar. Besides, GenImage employs state-of-the-art diffusion generators, such as Midjourney [20] and Stable Diffusion [25]. The richer image content and generator categories ensure the diversity of images in GenImage. Compared with general fake image detection datasets, GenImage also demonstrates the advantages of the dataset scale. We perform a comprehensive analysis of GenImage using existing SOTA detection methods and then propose two tasks resembling real-world detection problems: **(1) Cross-Generator Image Classification:** training the detectors on the images generated by one generator and evaluating the detectors on the images generated by the other generators. **(2) Degraded Image Classification:** evaluating the detectors on the degraded images, such as low resolution, JPEG compression, and Gaussian Blur. The GenImage is promising to significantly contribute to the advancement of detectors specifically designed for detecting AI-generated images.

## 2 Related Work

### 2.1 AI-generated Image Detection Datasets

The rapid growth of generative neural networks has facilitated the generation of images, raising concerns over the propagation of misinformation due to the low-cost production of such images. Generative networks primarily employ GANs for image generation, including CycleGAN [39], StyleGAN [13], ProGAN [12]. These generators produce images bearing strong resemblance to real images, although humans could often detect their synthetic nature. However, the recent advent of alternative generators, such as diffusion models, significantly enhances the quality of generated images, rendering the task of distinguishing real from fake images increasingly challenging. To aid in the development of detectors, initial synthetic image detection datasets [14, 16] predominantly

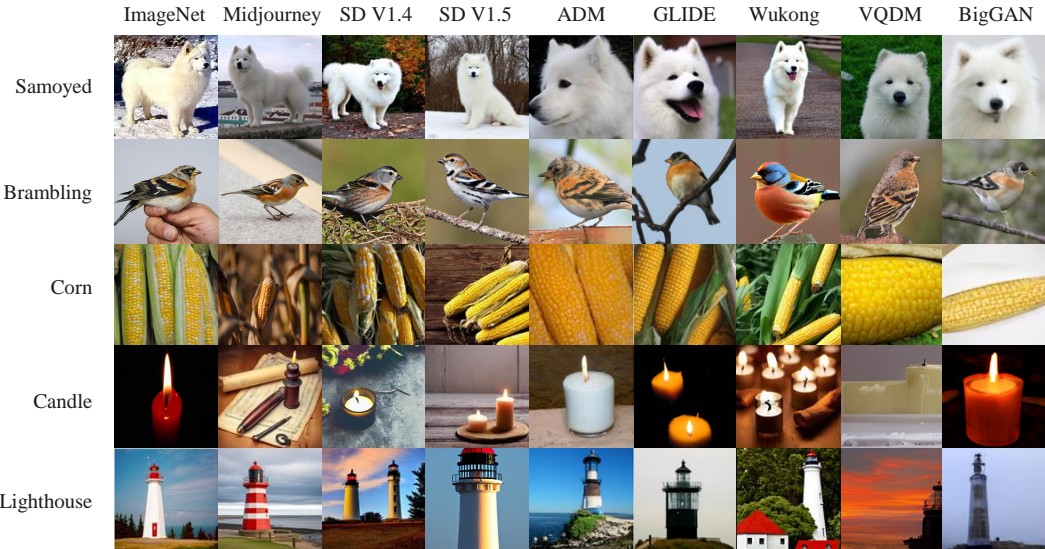

ImageNet  Midjourney  SD V1.4  SD V1.5  ADM  GLIDE  Wukong  VQDM  BigGAN

Samoyed

Brambling

Corn

Candle

Lighthouse

Figure 1: Visualization of images on GenImage dataset. SD is short for Stable Diffusion.

comprised a large number of facial images. The detectors based on these datasets cannot effectively identify the general images such as houses. Despite the limitations of early GANs in producing general images, pioneering work has made strides, such as CNNSpot [31], which solely employs ProGAN for training dataset generation and subsequently evaluates detector performance across a variety of GAN-based testing sets. A recent trend in this space involves the introduction of diffusion models as generators. For instance, DeepArt [32] leverages advanced generators such as Stable Diffusion, DALL-E2, Imagen, Midjourney, and Parti to produce art-like images. Similarly, initiatives like the IEEE VIP Cup [29] and DE-FAKE [27] have employed diffusion models to generate more general images. Based on a small-scale Cifar10 dataset [15], CiFAKE [1] generates fake images with Stable Diffusion V1.4. Nonetheless, these datasets often focus on GAN, and they are data-limited.

## 2.2 AI-generated Image Detection Methods

Numerous approaches have emerged in the pursuit of spotting generated images based on fake image detection datasets. Among the conventional methods, image classifiers have served as the most direct detectors, leveraging architectures like CNN-based ResNet50 [9], Transformer-based DeiT-S [28], and Swin-T [17] for binary classification of images. These detectors offer solid baseline techniques. In the early stages, emphasis was placed on detectors that relied on facial features. F3Net [23]] proposes to simultaneously explore frequency components partition and discrepancy of frequency statistics between real and fake images for face forgery detection. GramNet [18] uses global texture features to make fake face detection more robust and generalizable. These methods can, to some extent, be useful for the design of general image detectors. Past general image detectors have been developed and evaluated on images generated by GAN generators. Spec [38] takes the frequency spectrum as input and synthesizes GAN artifacts in real images without the necessity of specific GAN models to generate fake images as training data. DIRE [33] computes an error between the input and its reconstruction counterpart by a diffusion model.CNNSpot [31] uses ResNet-50 [9] as a binary classifier, with specific pre- and post-processing and data augmentation. Most of the detectors are for GAN. The development of dedicated detectors tailored to the unique characteristics of mixed GAN and diffusion data is imperative.

## 3 Dataset Construction

### 3.1 Dataset Details

To facilitate a precise evaluation of a detector's capacity to discern between AI-generated and real images, we construct a dataset, GenImage, comprising over one million pairs of real and fake images. Considering that ImageNet is an awesome dataset that we want to be on par with, GenImage employs

all the real images in ImageNet. Image generation in GenImage leverages 1000 distinct labels in ImageNet, ensuring a near-equal distribution of real and generated images across each class. Our dataset, GenImage, comprises 2,681,167 images, segregated into 1,331,167 real and 1,350,000 fake images. The real images are subdivided into 1,281,167 images for training and 50,000 for testing. With ImageNet [4] providing 1000 distinct image classes, we generate 1350 images for each class, out of which 1300 are allocated for training and the remaining 50 for testing.

To address the problem of detecting images generated by SOTA generators, we employ eight generative models for image generation, namely BigGAN [2], GLIDE [21], VQDM [8], Stable Diffusion V1.4 [25], Stable Diffusion V1.5 [25], ADM [5], Midjourney [20], and Wukong [35]. Each generator produces nearly the same number of images for each class, with 162 images for training and 6 for testing, with the exception of Stable Diffusion V1.5, which generates 166 images for training and 8 for testing. A combination of the fake images generated by a generator and their corresponding real images can be considered as a subset, such as Stable Diffusion V1.4 subset. The real images are not shared across subsets. The number of generated images approximates the number of real images for the whole dataset and for each subset. The almost equal and high number of images generated by each generator in this dataset allows the properties of each generator to be fully explored when developing detection models without being affected by the imbalances in numbers. Our model input sentences follow the template "photo of class", with "class" being substituted by ImageNet labels. For Wukong, Chinese sentences often achieve better generation quality. Thus, the sentences are translated into Chinese for the web API. For ADM and BIGGAN, we employ their pre-trained models on ImageNet, inputting the labels to generate images.

The generated images are shown in Figure 1. More visualized images are provided in the supplementary materials. It can be observed that overall the generated images are similar to the real images in ImageNet. Images with the same labels are analyzed in more detail. Animals and plants succeed in keeping the appearance of the target object consistent, e.g., samoyed with a similar appearance, and they differ in movement, perspective, and background. The appearance of the targets also varies for objects such as candles and lighthouses. Therefore, the generated images have a high degree of variability and reasonableness.

## 3.2 Fake Image Generators

**Diffusion Model** has recently achieved remarkable performances in image synthesis. Midjourney [20] is one of the most renowned commercial software programs, known for its exceptional image generation performance. We utilize Midjourney V5 for image generation, which offers more intricate details compared to previous versions, resulting in images that closely resemble real-world photographs. The resolution of images generated by Midjourney is $1024 \times 1024$. Wukong [35] is a large-scale text-to-image generative model based on the diffusion model. This model is trained on the largest Chinese open-source multimodal dataset, the Wukong dataset, making it particularly suitable for Chinese language processing. The image resolution is $512 \times 512$. Stable Diffusion [25] is an advanced text-to-image diffusion model capable of generating highly realistic images based on any given text input. Stable Diffusion V1.4 is pretrained from the Stable Diffusion V1.2 checkpoint and fine-tuned on 225k steps at resolution 512x512 on Laion-Aesthetics V2 5+ dataset and drops 10% of the text-conditioning. The training settings of Stable Diffusion V1.5 is the same as Stable Diffusion V1.4, except that Stable Diffusion V1.5 is fine-tuned on 595,000 steps. Both Stable Diffusion V1.4 and Stable Diffusion V1.5 generate images with a resolution of $512 \times 512$. ADM [5] proposes a diffusion model which achieves better sample quality than GANs. We use a model with classifier guidance, which is pretrained on ImageNet. GLIDE [21] is a diffusion model for text-conditional image synthesis. GLIDE uses a text encoder to train a 3.5 billion diffusion model. We use classifier-free guidance for generating images. Its resolution is $256 \times 256$. VQDM [8] proposes a latent-space method that eliminates the undirectional bias with previous methods and incorporates a mask-and-replace diffusion mechanism to alleviate the accumulation of errors. The resolution is $256 \times 256$.

**GAN** has brought significant quality improvements in image generation in the past decades. BigGAN [2] is a representative method in GAN family. BigGAN makes three contributions, including architectural changes, sampling techniques, and instabilities reduction techniques. We use the BigGAN model pretrained on ImageNet. The image resolution is $128 \times 128$.

Table 2: Results of cross-validation on different training and testing subsets using ResNet-50.

| Training Subset | Testing Subset | | | | | | | | Avg Acc.(%) |
|---|---|---|---|---|---|---|---|---|---|
| | Midjourney | SD V1.4 | SD V1.5 | ADM | GLIDE | Wukong | VQDM | BigGAN | |
| Midjourney | **98.8** | 76.4 | 76.9 | 64.1 | 78.9 | 71.4 | 52.5 | 50.1 | 71.1 |
| SD V1.4 | 54.9 | **99.9** | 99.7 | 53.5 | 61.9 | 98.2 | 56.6 | 52.0 | **72.1** |
| SD V1.5 | 54.4 | 99.8 | **99.9** | 52.7 | 60.1 | 98.5 | 56.9 | 51.3 | 71.7 |
| ADM | 58.6 | 53.1 | 53.2 | **99.0** | 97.1 | 53.0 | 61.5 | 88.3 | 70.4 |
| GLIDE | 50.7 | 50.0 | 50.1 | 56.0 | **99.9** | 50.3 | 51.0 | 74.0 | 60.2 |
| Wukong | 54.5 | 99.7 | 99.6 | 51.4 | 58.3 | **99.9** | 58.7 | 50.9 | 71.6 |
| VQDM | 50.1 | 50.0 | 50.0 | 50.7 | 60.1 | 50.2 | **99.9** | 66.8 | 59.7 |
| BigGAN | 49.9 | 49.9 | 49.9 | 50.6 | 68.4 | 49.9 | 50.6 | **99.9** | 58.6 |

## 4 GenImage Benchmark

### 4.1 Fake Image Detectors

In order to evaluate our dataset, we investigate and select some existing fake image detection methods.

**Backbone Model** can be directly utilized as the detector for the binary classification of real and fake images. We use ResNet-50 [9], DeiT-S [28] and Swin-T [17] as the fake image detector. ResNet [9] is based on convolutional neural network. DeiT-S [28] and Swin-T [17] are based on Transformer. Without a specific design for the fake image detection task, the backbone models can be considered baseline methods.

**Fake Face Detector** has been developed for a long time. F3Net [23] proposes to simultaneously explore frequency components partition and discrepancy of frequency statistics between real and fake images for face forgery detection. GramNet [18] uses global texture features to make fake face detection more robust and generalizable. These methods train models on face images. These models are difficult to directly work well with images beyond the face domain. However, these methods can still inspire the design of general image detectors.

**General Fake Image Detector** employs special designs for classifying general images getting rid of the limitation of face content. Spec [38] takes the frequency spectrum as input and synthesizes GAN artifacts in real images without the necessity of specific GAN models to generate fake images as training data. CNNSpot [31] uses ResNet-50 as a binary classifier, with specific pre- and post-processing and data augmentation. However, the performance of existing methods on datasets comprising a combination of GAN and diffusion-generated images requires further improvement. Consequently, the development of detectors tailored to the unique characteristics of mixed GAN and diffusion data becomes imperative.

### 4.2 Task 1: Cross-Generator Image Classification

We first evaluate the performance of the detector when trained and tested on images generated by the same generator. We perform our analysis with the most commonly used Resnet-50 [9]. Our GenImage dataset consists of eight distinct subsets, each corresponding to a specific generator. Within each subset, we further divide the data into training and testing sets, each comprising 1000 classes of images. As shown in Table 2, training and testing within each subset consistently yield accuracy rates surpassing 98.5%. Notably, the Stable Diffusion V1.4 and Stable Diffusion V1.5 subsets achieve an exceptional accuracy of 99.9%. However, we observe a substantial performance degradation when training and testing are conducted using different generators. For instance, when the ResNet-50 is trained on Stable Diffusion V1.4 and tested on Midjourney, the binary classification accuracy drops to 54.9%.

Based on this observation, detecting a fake image synthesized by a specific generator is relatively straightforward, simply by training the binary classifier on a dataset consisting of real images and fake images. However, this approach is likely to be tied to this generator and will not perform well on unknown generators. In real-world scenarios, the generator is often unknown at training. In this work, we hope to be able to effectively evaluate the generalization ability of the detector, i.e. the ability to distinguish between real and fake images independent of the used generator. We propose the cross-generator image classification task to evaluate the recognition ability of the AI-generator detector. In Table 2, training on Stable Diffusion V1.4 achieves the best results. To this end, we train the model on Stable Diffusion V1.4 and subsequently test it on the testing subsets from different

Table 3: Results of different methods trained on SD V1.4 and evaluated on different testing subsets.

| Method | Testing Subset | | | | | | | | Avg Acc.(%) |
|---|---|---|---|---|---|---|---|---|---|
| | Midjourney | SD V1.4 | SD V1.5 | ADM | GLIDE | Wukong | VQDM | BigGAN | |
| ResNet-50 [9] | 54.9 | **99.9** | 99.7 | **53.5** | 61.9 | 98.2 | 56.6 | 52.0 | 72.1 |
| DeiT-S [28] | 55.6 | **99.9** | 99.8 | 49.8 | 58.1 | 98.9 | 56.9 | 53.5 | 71.6 |
| Swin-T [17] | **62.1** | **99.9** | 99.8 | 49.8 | **67.6** | 99.1 | **62.3** | **57.6** | **74.8** |
| CNNSpot [31] | 52.8 | 96.3 | 95.9 | 50.1 | 39.8 | 78.6 | 53.4 | 46.8 | 64.2 |
| Spec [38] | 52.0 | 99.4 | 99.2 | 49.7 | 49.8 | 94.8 | 55.6 | 49.8 | 68.8 |
| F3Net [23] | 50.1 | **99.9** | **99.9** | 49.9 | 50.0 | **99.9** | 49.9 | 49.9 | 68.7 |
| GramNet [18] | 54.2 | 99.2 | 99.1 | 50.3 | 54.6 | 98.9 | 50.8 | 51.7 | 69.9 |

generators in this task. We then compute the average accuracy across the various test subsets, as shown in Table 3. In Table 3, for each method, we train one model on one generator and evaluate this model on eight generators. We average eight results for one method. In Table 4, for each method, we train eight models on eight generators respectively. We evaluate each model on eight generators. We average sixty-four results for one method. For example, we respectively train eight ResNet-50 models using eight generators and average their evaluation results on Midjourney, yielding an accuracy of 59.0%. The evaluation is also performed on the other generators, such as Stable Diffusion V1.4, which leads to another fifty-six evaluation results. All the sixty-four testing results are then averaged to achieve 66.9%. In summary, the cross-generator training and evaluation contains two settings. The first setting is training a model on SD V1.4, and evaluating the eight generators. The second setting is the same as the training and evaluation in Table 2. We average the eight average accuracies in Table 2 to obtain the results of ResNet-50 in Table 4. Our evaluation offers comprehensive insights into the capabilities of the detectors. We conduct a thorough evaluation of various backbone architectures on our dataset, including the CNN-based ResNet50 [9], as well as the Transformer-based DeiT-S [28] and Swin-T [17]. This evaluation aims to assess the performance and effectiveness of these architectures in AI-generated image detection task. Additionally, we also evaluate the performance of exsiting AI-generated image detectors, such as CNNSpot [31] and Spec [38].

In Table 3 and Table 4, current CNN-based models like ResNet perform similarly to Transformer-based models like DeiT-S and Swin-T. These backbone models share similar computation costs and parameters, and they are often used for 1000-class image classification on ImageNet. For ImageNet 1000 class image classification, the backbone models tend to focus more on the classification of the content of the image. For GenImage, the focus is more on the pattern of discriminating between real and fake images. We train them from scratch for binary classification. In the cross-generator image classification task, the best result is achieved by Swin-T, while the other two methods are close in accuracy. Improving Transformer-based methods is a promising direction in our dataset. We also evaluate the existing AI-generated image detectors, e.g., CNNSpot [31], and Spec [38]. We train these models on our dataset and evaluate them. CNNSpot proposes that the recognition performance can be improved by augmenting the training data: (1) Images are blurred with $\sigma \sim$ Uniform[0,3] with 50% probability. (2) Images are JPEG-ed with 50% probability. GAN often causes unique artifacts in the up-sampling component. Spec show that such artifacts are manifested as replications of spectra in the frequency domain. Thus, Spec takes spectrum instead of pixel as its input for classification. CNNSpot and Spec work well on their GAN-based datasets. However, they perform even worse than the baseline backbone models on our dataset, mainly composed of images generated by the diffusion model. F3Net [23] contains two branches, namely Frequency-aware Image Decomposition (FAD) and Local Frequency Statistics (LFS). FAD studies manipulation patterns through frequency-aware image decomposition. LFS extracts local frequency statistics. Besides, a mixed block is utilized for collaborative feature interaction. GramNet [18] observes the difference between fake and real faces. Then it leverages global texture features to enhance robustness and generalization. Both methods are inferior to training directly with a binary classification backbone neural network model. Considering the advantages of all these approaches, designing a special backbone network could be a promising solution for GenImage.

## 4.3 Task 2: Degraded Image Classification

Images often encounter degradation problems during propagation [31], such as low resolution, compression, and noise interference. Detectors are supposed to be robust to these challenges. To address this, we propose evaluating the performance of the detector on these degraded images, which

Table 4: Results of cross-validation on different training and test subsets using different methods. Eight models trained on eight generators are tested on one generator, and their average accuracy is each data point in the testing subset column.

| Method | Testing Subset | | | | | | | | Avg Acc.(%) |
|---|---|---|---|---|---|---|---|---|---|
| | Midjourney | SD V1.4 | SD V1.5 | ADM | GLIDE | Wukong | VQDM | BigGAN | |
| ResNet-50 [9] | 59.0 | 72.3 | 72.4 | 59.7 | 73.1 | 71.4 | 60.9 | 66.6 | 66.9 |
| DeiT-S [28] | 60.7 | 74.2 | 74.2 | 59.5 | 71.1 | 73.1 | 61.7 | 66.3 | 67.6 |
| Swin-T [17] | **61.7** | **76.0** | **76.1** | 61.3 | **76.9** | **75.1** | **65.8** | **69.5** | **70.3** |
| CNNSpot [31] | 58.2 | 70.3 | 70.2 | 57.0 | 57.1 | 67.7 | 56.7 | 56.6 | 61.7 |
| Spec [38] | 56.7 | 72.4 | 72.3 | 57.9 | 65.4 | 70.3 | 61.7 | 64.3 | 65.1 |
| F3Net [23] | 55.1 | 73.1 | 73.1 | **66.5** | 57.8 | 72.3 | 62.1 | 56.5 | 64.6 |
| GramNet [18] | 58.1 | 72.8 | 72.7 | 58.7 | 65.3 | 71.3 | 57.8 | 61.2 | 64.7 |

Table 5: Model evaluation on degraded images. q denotes quality.

| Method | Testing Subset | | | | | | Avg Acc(%) |
|---|---|---|---|---|---|---|---|
| | LR (112) | LR (64) | JPEG (q=65) | JPEG (q=30) | Blur ($\sigma$=3) | Blur ($\sigma$=5) | |
| ResNet-50 [9] | 96.2 | 57.4 | 51.9 | 51.2 | **97.9** | 69.4 | 70.6 |
| DeiT-S [28] | 97.1 | 54.0 | 55.6 | 50.5 | 94.4 | 67.2 | 69.8 |
| Swin-T [17] | 97.4 | 54.6 | 52.5 | 50.9 | 94.5 | 52.5 | 67.0 |
| CNNSpot [31] | 50.0 | 50.0 | **97.3** | **97.3** | 97.4 | 77.9 | 78.3 |
| Spec [38] | 50.0 | 49.9 | 50.8 | 50.4 | 49.9 | 49.9 | 50.1 |
| F3Net [23] | 50.0 | 50.0 | 89.0 | 74.4 | 57.9 | 51.7 | 62.1 |
| GramNet [18] | **98.8** | **94.9** | 68.8 | 53.4 | 95.9 | **81.6** | **82.2** |

more accurately simulate practical conditions, as shown in Table 5. After training the detector on our Stable Diffusion V1.4 subset, we employ a series of methods to degrade only the testing set images. Specifically, we downsample the images to resolutions of 112 and 64 only in the testing set. Furthermore, we utilize JPEG compression with quality ratios of 65 and 30, introducing compression artifacts. To introduce blurring effects, we incorporate Gaussian Blurring. As baseline models, ResNet-50, DeiT-S, and Swin-T all present similar results. They all detect images well for the low resolution at $112 \times 112$. However, for the lower resolution of $64 \times 64$, they present poorer results. The experimental results suggest that these models are more sensitive to resolution. For Gaussian blurring, these models also demonstrate similar performance. JPEG compression has a significant impact on these backbone models. It is worth noting that CNNSpot is robust to both JPEG compression and Gaussian blurring, which is basically due to the fact that CNNSpot uses JPEG compression and Gaussian blurring as additional data preprocessing during training. Therefore, designing a reasonable preprocessing method is a promising way to solve the degraded image classification problem in our dataset. By applying these detectors to degraded images, we gain valuable insights into their performance under various challenging conditions. This evaluation provides a more comprehensive understanding of the capability of the detector to handle practical image degradation.

## 5 GenImage Analysis

In this section, we conduct extensive experiments to investigate the characteristics of GenImage, and the effectiveness of this dataset.

### 5.1 Effect of Increasing the Number of Images

In general, increasing the scale of the dataset has been commonly believed to improve the performance of the classification model. However, this observation has not been fully explored in the fake image detection task. To assist the assessment of necessary data quantity, we conduct comprehensive experiments, as shown in Table 6. We conduct a preliminary experiment using Stable Diffusion V1.4 generated images. In particular, our training set is generated solely by Stable Diffusion V1.4. Two different settings are introduced in the testing set. In identical-generator setting, the testing set is composed of 6000 fake images. On the other hand, in the cross-generator setting, the testing set contains 50000 fake images from 8 different generators. The same number of real images is collected from ImageNet [4]. Firstly, we train a ResNet-50 on a dataset consisting of 10 classes, with each class containing 160 fake images. The number of real images is identical to the number of fake

Table 6: Results of increasing the number of images.

| Total | Image Classes | Images Per Class | Identical-Generator Acc. (%) | Cross-Generator Acc. (%) |
|---|---|---|---|---|
| 1600 | 10 | 160 | 73.6 | 60.9 |
| 1600 | 100 | 16 | 81.8 | 62.7 |
| 16000 | 100 | 160 | 97.7 | 68.7 |
| 16000 | 1000 | 16 | 96.3 | 68.4 |
| 162000 | 1000 | 162 | 99.9 | 72.1 |

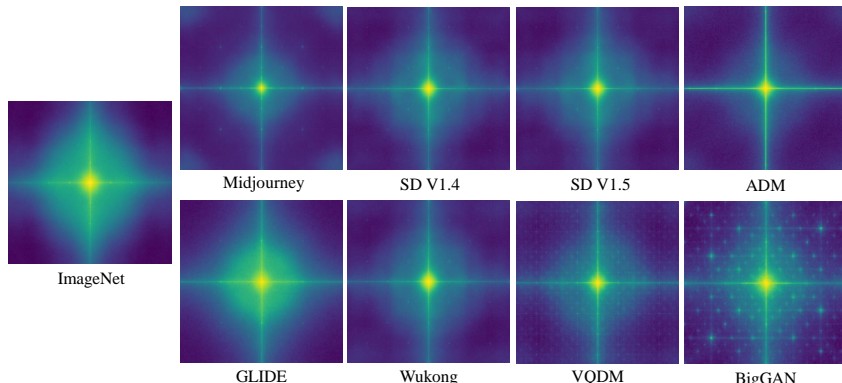

Figure 2: Frequency analysis on real images and generated images.

images. The accuracy achieved on the identical-generator testing set was only 73.6%, and even lower on the cross-generator testing set. We further expand our training set from two aspects: by increasing the number of classes and by increasing the number of images within each class. Encouragingly, we observe that the performance substantially improves with both approaches. This observation underscores the significance of augmenting the training set.

## 5.2 Frequency Analysis

In Figure 2, we visualize the average spectral spectra of real images and fake images from different generators. Following CNNSpot [31], we use the discrete Fourier transform to investigate artifacts generated by the generative model. The noise residuals $R$ can be computed by the formula $R = X - F(X)$, where $X$ is the input image, and $F(\cdot)$ is the denoising filter DCNN [37]. For each image source, we average the noise residuals of 1000 randomly selected images and use the Fourier transform of the results for spectral analysis. Comparing the real image, the image generated by the GAN, and the image generated by the Diffusion model, some interesting results can be discovered. The results show that for GAN, artifacts are shown in the form of a regular grid. Real images from ImageNet contain few artifacts, as well as diffusion models. These results reflect that images from the diffusion models are closer to the real image than BigGAN, and therefore diffusion models present a greater challenge for detection. Durall et al. [6] show that the up-sampling methods, such as the up-convolution or transposed convolution lead to the inability of GANs to correctly approximate the spectral distributions of training data, which explains why the artifacts occur in GANs. Richer et al. [24] observe that the diffusion models do not produce grid-like artifacts in the frequency spectrum, but exhibit a systematic mismatch towards higher frequencies. The hypothesis is that less weight is attached to the higher frequencies during training since matching lower frequencies is more important to the perceived quality of generated images.

## 5.3 Image Class Generalization

In order to verify that the detector trained on our dataset can well generalize to different contents of images, we sample three subsets of classes, i.e., 10, 50, and 100, for training ResNet-50. It can be seen that it is possible to generalize well to other classes of images using only a subset of the 1000 classes, and the more classes in the subset, the better the generalization performance. We use all the generators for training and testing. We first keep the number of training data constant, i.e., 12,800 images are generated. The testing set contains 1000 classes of images with 50 generated

images for each class. The training set and the testing set encompass the images generated by the eight generators. The number of real images in each class is identical to that of the fake images.

As shown in Table 7, training on only a subset of image classes can also achieve good results on 1000 classes of images, and more classes of images achieve higher accuracy. This result validates the generalization performance of our dataset over image classes, which also implies training on 1000 classes can achieve better results on unseen images. We also compare the accuracy under two settings: 100 classes of images with 128 images for each class and 100 classes of images with 1300 images for each class. It can be seen that increasing the number of images for each class improves the accuracy.

Table 7: Results of increasing the class.

| Image Classes | Fake Images Per Class | Acc.(%) |
|---|---|---|
| 10 | 1280 | 78.8 |
| 50 | 256 | 80.3 |
| 100 | 128 | 81.2 |
| 100 | 1300 | 98.4 |

## 5.4 Generator Correlation Analysis

As the generator category is not commonly available as prior knowledge, we conduct an extensive evaluation on the cross-generator image classification task, encompassing eight different generative methods, as shown in Figure 3. ResNet-50 is the detector. It can be seen that the same generator achieves optimal performance in both training and testing. Generators exhibiting higher similarity tend to yield superior cross-generator performance. For instance, Stable Diffusion V1.4 and Stable Diffusion V1.5, sharing similar architecture, demonstrate good classification results when trained on the former and tested on the latter. Likewise, the detection model trained on Stable Diffusion models can effectively generalize to Wukong. From Figure 3 (a), we can see that training on Stable Diffusion V1.4, Stable Diffusion V1.5, and Wukong yields the best overall generalization performance. Midjourney poses the greatest challenge in terms of generalization. Figure 3 (a) is the accuracy score map, which shows the accuracies with different training and testing approaches. In Figure 3 (b), we compute Pearson correlation based on the accuracy score map. Pearson correlation coefficient is a measure of linear correlation between two sets of data. Figure 3 (b) highlights that models with similar architectures tend to exhibit higher relevance, such as Stable Diffusion V1.4, Stable Diffusion V1.5, and Wukong.

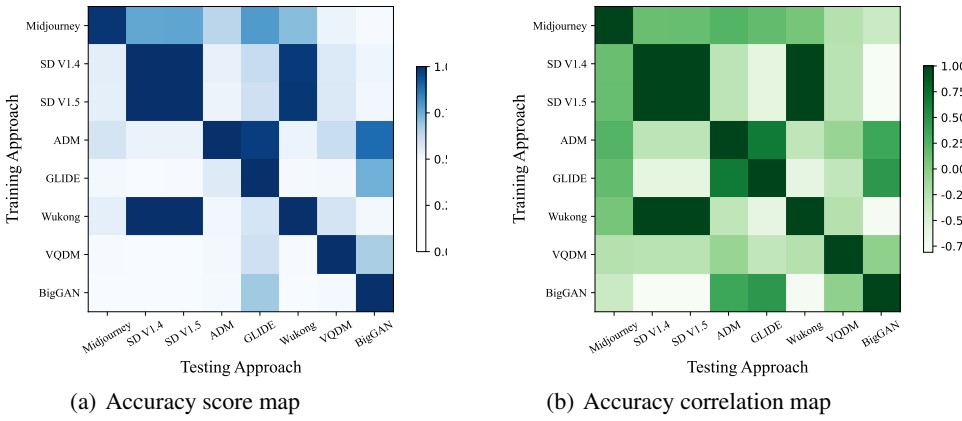

(a) Accuracy score map      (b) Accuracy correlation map

Figure 3: A visualization of generator correlations.

## 5.5 Image Content Generalization

Our dataset encompasses a broad range of image classes, extending beyond the conventional limitations of face and art images. As demonstrated in Table 8, our dataset exhibits notable content generalization capability. Specifically, a ResNet-50 trained on our dataset effectively works on face and art images. LFW [11] is a publicly available database of face images designed for face recognition problem. We have assembled 10,000 face images from the LFW dataset, complemented by an equal number of images generated using face labels from the same dataset. Laion-Art is a subset of Laion-5B [26]. Several lightweight models estimate how people would rate each image in Laion-5B based on aesthetics, and the images with high scores are kept for Laion-Art.

DiffusionDB [34] is a large-scale gallery dataset with 1.8 million unique prompts. We have accumulated 10,000 art images from Laion-Art, alongside 10,000 art images generated using prompts from DiffusionDB, as shown in Figure 4. For training, we employ the Stable Diffusion V1.4 subset of GenImage. Stable Diffusion V1.4 is also used to generate the face and art fake images. We directly evaluate our model on the testing dataset without finetuning. As shown by the table, our dataset exhibits a commendable generalization capacity, achieving over 95.0% accuracy in identifying face and art images.

Table 8: Image content generalization.

| Image | Image Source | | Acc. |
|---|---|---|---|
| Content | Real | Fake | (%) |
| Face | LFW | SD V1.4 | 99.9 |
| Art | Laion-Art | SD V1.4 | 95.0 |

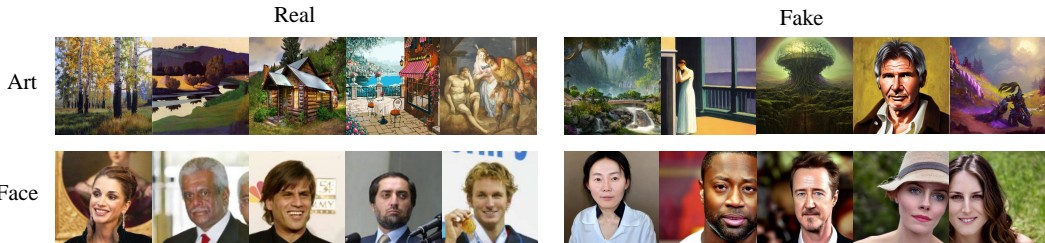

Figure 4: The visualization of the face images and art images.

## 6 Conclusion

This paper introduces GenImage, a dataset specifically designed for detecting fake images generated by generative models. GenImage serves as a million-scale benchmark, surpassing previous datasets and benchmarks in terms of the number of images, image content, and generator selection. We propose two tasks, namely cross-generator image classification and degraded image classification, to assess the detection performance of existing detectors on GenImage. Additionally, a detailed analysis of the dataset is provided, enabling us to gain insights into how GenImage can contribute to the development of detectors for real-world applications.

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
