# OpenReview forum: "GenImage: A Million-Scale Benchmark for Detecting AI-Generated Image"
_NeurIPS.cc/2023/Track/Datasets_and_Benchmarks — NeurIPS 2023 Datasets and Benchmarks Poster_

### Official Review · Reviewer_ZWev · 2023-07-20

**Rating:** 9
**Confidence:** 5
**Correctness:** the paper is technically correct.
**Clarity:** the paper is well written and well or…

**Strengths:**

 The dataset addresses an important topic with potentially brought impact and closes the large gap to existing benchmarks which where mostly missing diverse data from Diffusion models. The provided analysis shows, that in contrast to GAN data, current detection models fail to transfer from and to Diffusion Models. this makes it even more urgent for the community to have such a large and diverse benchmark.

**Additional Feedback:**

In order to establish this benchmark the authors should think about adding an additional hidden testset and maintain a leader board. this is a lot of work, but would provide the best independent evaluation of novel detection methods...

**Documentation:**

The dataset is well documented and looks ready to use with a low entrance barrier

**Ethics:**

no ethical concerns

**Limitations:**

the generation of images by single classes mostly produces images with one central object (=class lable). This could introduce some limitation of generality as more complex scenes are not present. especially for the high-res images, it would be possible to generate more complex, multi object content...

**Opportunities For Improvement:**

A) the related work and the selection of baseline algorithms is missing some of the latest approaches. This is not to be held against this submission, since some have not been published at the time of submission and are only available on arxiv. However, given the expected advance in transferability, it would be wise to add them in a final version to have a baseline which is not outdated from the start.

Example References:
DIRE: https://arxiv.org/abs/2303.09295

B) section 4.2 (Frequency Analysis)
here the authors should cite [1], as this paper actually explains why these artifact occur in GANs. Also you might want to consider to user their 1D visualization of spectra for fig 2 as the 2d spectra are very hard to interpret.

Also, that  fact that diffusion Models are not having these artifact has been shown in [2] already

[1] Durall, R., Keuper, M., & Keuper, J. (2020). Watch your up-convolution: Cnn based generative deep neural networks are failing to reproduce spectral distributions. In Proceedings of the IEEE/CVF conference on computer vision and pattern recognition (pp. 7890-7899).

[2]  https://arxiv.org/abs/2210.14571

**Relation To Prior Work:**

only minor references missing - see improvements

**Summary And Contributions:**

The paper introduces a large dataset for the evaluation of detection algorithms for generated images. With  ~2.6M annotated images (1.3M real and 1.3M synthetic), generated  by a wide range of SOTA GAN and Diffusion models, the proposed dataset is currently the largest and most diverse benchmark in this field. The additionally provided evaluation of baseline detection methods, including ablations towards sample size, compression and blurring provides a good starting ground to establish this as a generally accepted standard evaluation benchmark.

---

> ### Author Response · Authors · 2023-08-20
> **Responses to Reviewer ZWev**
>
> Thanks to the reviewer for the valuable comments. We respond to the questions in the following.
> > **Q1.** The related work and the selection of baseline algorithms is missing some of the latest approaches. This is not to be held against this submission, since some have not been published at the time of submission and are only available on arxiv. However, given the expected advance in transferability, it would be wise to add them in a final version to have a baseline which is not outdated from the start.  Example References: DIRE: https://arxiv.org/abs/2303.09295
>
> * **A1.** Thanks for the nice suggestion. We will cite DIRE in the final version. However, this paper has not released code in its GitHub repository, which makes it hard to reimplement the method. If this paper releases codes, we will consider adding them in the final version as a baseline method.
>
> > **Q2.** Section 4.2 (Frequency Analysis) here the authors should cite [1], as this paper actually explains why these artifact occur in GANs. Also you might want to consider to user their 1D visualization of spectra for fig 2 as the 2d spectra are very hard to interpret. Also, that fact that diffusion Models are not having these artifact has been shown in [2] already.
>
> * **A2.** Thanks for the suggestion. We will cite these two papers and add an explanation of why the artifacts occur in GANs. We will add the fact that the artifacts do not occur in Diffusion models. Durall et al. [1] show that the up-sampling methods, such as the up-convolution or transposed convolution lead to the inability of GANs to correctly approximate the spectral distributions of training data. Richer et al. [2] observe that the diffusion models do not produce grid-like artifacts in the frequency spectrum, but exhibit a systematic mismatch towards higher frequencies. The hypothesis is that less weight is attached to the higher frequencies during training since matching lower frequencies is more important to the perceived quality of generated images.
>
> [1] "Watch your up-convolution: Cnn based generative deep neural networks are failing to reproduce spectral distributions." CVPR 2020.
>
> [2] "Towards the detection of diffusion model deepfakes." ArXiv 2022.
>
> > **Q3.** the generation of images by single classes mostly produces images with one central object (=class lable). This could introduce some limitation of generality as more complex scenes are not present. especially for the high-res images, it would be possible to generate more complex, multi object content...
>
> * **A3.** The generalization performance of GenImage has been evaluated on Face and Art Images. We further conduct experiments on images generated by richer text descriptions and image-to-image generation, as shown in the following Table. To obtain richer text descriptions, we collect 1000 prompts and 1000 images from CC12M [3]. For obtaining multi-object content images, we use a template of "photo of {class A}, {class B}, and {class C}". The classes and real images come from ImageNet. For an image-to-image generation, we input the real images from ImageNet and a template of "a painting of {class}" to the generator. The generator used in the above experiments is Stable Diffusion.
>
> |           | Richer Text Descriptions | Image-to-Image Generation | Multi-Object Content |
> |:-:|:-:|:-:|:-:|
> | ResNet-50 | 99.9                     | 99.3                      | 99.9                 |
>
>
> [3] "Conceptual 12m: Pushing web-scale image-text pre-training to recognize long-tail visual concepts." CVPR 2021.
>
> > **Q4.** In order to establish this benchmark the authors should think about adding an additional hidden testset and maintain a leader board. this is a lot of work, but would provide the best independent evaluation of novel detection methods...
>
> * **A4.** Thanks. We will add an additional hidden test set with more than 50000 images and maintain a leader board. The test set will be uploaded to the GitHub Page.

---

> > ### Comment · Reviewer_ZWev · 2023-08-21
> > **Paper revision**
> >
> > Thank you for the response. All of my points as well as the questions and comments by the other reviews have been addressed sufficiently in my opinion. I will keep my high rating and will argue in favor of this paper. However, there are two things the authors could provide to aid the upcoming discussions: according to the official author rebuttal mail you can also:
> >
> > * *You can make revisions to your paper and supplementary materials, and you are allowed **one additional page** to address the reviewers’ comments. Ensure that it is easy for reviewers to find how their comments were addressed.*
> >
> > * *You can respond to each review with a separate response, and also have a “global” response to all reviewers jointly. Use the “Official Comment” button to submit these.*
> >
> > I would highly recommend to make use of these options. Please color-code changes in the paper and summarize them in a global response.

---

> > > ### Author Response · Authors · 2023-08-22
> > > **Responses to Reviewer ZWev - Part 2**
> > >
> > > Thanks for your useful suggestions. We have revised the submitted main paper and supplemental materials based on the reviewers' suggestions, and we mark the changes in red. We also have a global response to all reviewers.

---

### Official Review · Reviewer_GjRx · 2023-07-21
**Marginally above acceptance threshold**

**Rating:** 6
**Confidence:** 2
**Correctness:** yes
**Clarity:** good

**Strengths:**


The paper makes some significant contributions to the field of fake image detection, including:

The construction of a large-scale benchmark dataset with a diverse range of image categories and state-of-the-art generative models, which can facilitate the development and evaluation of fake image detectors.

The proposal of two evaluation tasks, including cross-generator image classification and degraded image classification, which can simulate real-world scenarios and provide a more comprehensive evaluation of fake image detectors.

The comprehensive analysis of the dataset using existing state-of-the-art fake image detectors, which can serve as a baseline for future research.

**Additional Feedback:**

no

**Documentation:**

yes

**Ethics:**

no ethics issue

**Limitations:**



The paper does not provide a detailed analysis of the performance of different generative models in generating fake images. It is unclear how the performance of the proposed dataset would be affected if different generative models were used.

The paper only evaluates the performance of existing fake image detectors on the proposed dataset, without proposing any new fake image detection methods. It would be interesting to see if the proposed dataset can inspire the development of more effective fake image detectors.

The paper does not provide a detailed discussion of the limitations of the proposed dataset. For example, it is unclear how the proposed dataset would perform in detecting fake images generated by new or unknown generative models.

**Opportunities For Improvement:**

more experiments on various methods need to be done in the future

**Relation To Prior Work:**

not very clear

**Summary And Contributions:**

The paper presents a new benchmark dataset called GenImage, which contains over one million pairs of real and AI-generated fake images across a diverse range of image categories. The authors use state-of-the-art generative models, including diffusion models and GANs, to generate the fake images. They propose two evaluation tasks, including cross-generator image classification and degraded image classification, and demonstrate the effectiveness of the dataset in evaluating the performance of existing fake image detectors.

Novelties:

The paper makes some significant contributions to the field of fake image detection, including:

The construction of a large-scale benchmark dataset with a diverse range of image categories and state-of-the-art generative models, which can facilitate the development and evaluation of fake image detectors.

The proposal of two evaluation tasks, including cross-generator image classification and degraded image classification, which can simulate real-world scenarios and provide a more comprehensive evaluation of fake image detectors.

The comprehensive analysis of the dataset using existing state-of-the-art fake image detectors, which can serve as a baseline for future research.
Pitfalls:

Although the paper presents some significant contributions, there are also some potential pitfalls :

The paper does not provide a detailed analysis of the performance of different generative models in generating fake images. It is unclear how the performance of the proposed dataset would be affected if different generative models were used.

The paper only evaluates the performance of existing fake image detectors on the proposed dataset, without proposing any new fake image detection methods. It would be interesting to see if the proposed dataset can inspire the development of more effective fake image detectors.

The paper does not provide a detailed discussion of the limitations of the proposed dataset. For example, it is unclear how the proposed dataset would perform in detecting fake images generated by new or unknown generative models.

---

> ### Author Response · Authors · 2023-08-20
> **Responses to Reviewer GjRx**
>
> Thanks to the reviewer for the valuable comments. We respond to the questions in the following.
> > **Q1.** It is unclear how the performance of the proposed dataset would be affected if different generative models were used.
>
> * **A1.** Thanks. The generalization performance of different generative models can be evaluated by the average accuracy in Table 2. For example, taking ResNet-50 as a standard detector, the average accuracy of Midjourney is 71.1%, which is higher than GLIDE (60.2%). The generalization performance of Midjourney can be considered better than GLIDE.
>
> > **Q2.** It would be interesting to see if the proposed dataset can inspire the development of more effective fake image detectors.
>
> * **A2.** Thanks for the suggestion. In this paper, we evaluate the existing detectors on GenImage and analyze their results in section 3.2 and section 3.3. Based on the analysis, the proposed dataset can inspire the development of new detectors in two directions. The first direction is designing a new backbone network. The second direction is designing a reasonable image preprocessing method.
>
> > **Q3.** It is unclear how the proposed dataset would perform in detecting fake images generated by new or unknown generative models.
>
> * **A3.** We conduct experiments to show that the models trained on our eight generators are sufficient to perform well on new or unknown generative models, as shown in the following Table. We demonstrate the ResNet-50 that is trained on all generators and evaluated on NVAE [1], CogView2 [2], StyleGAN [3], and IF [4]. For each generator, we collect 1000 real images and generate 1000 fake images. For CogView and IF, we use the images from ImageNet and the input sentences follow the template "photo of class", with "class" being substituted by ImageNet labels. We use an NVAE model pre-trained on FFHQ [5] to generate fake images, and the real images come from FFHQ. StyleGAN is pre-trained on LSUN bedroom, and the real images also come from LSUN [6]. The results show that the detector trained on our dataset can generalize to other kinds of generators. This discussion will be added to the supplementary material.
>
> |           | NVAE | CogView2 | StyleGAN | IF   |
> |:-:|:-:|:-:|:-:|:-:|
> | ResNet-50 | 93.4 | 97.5     | 97.9     | 90.2 |
>
> [1] "NVAE: A deep hierarchical variational autoencoder."NIPS 2020.
>
> [2] "Cogview2: Faster and better text-to-image generation via hierarchical transformers." NIPS 2022.
>
> [3] "A style-based generator architecture for generative adversarial networks." CVPR 2019.
>
> [4] https://github.com/deep-floyd/IF/tree/develop. 2023
>
> [5] "A style-based generator architecture for generative adversarial networks." CVPR 2019.
>
> [6] "Lsun: Construction of a large-scale image dataset using deep learning with humans in the loop." ArXiv 2015.

---

> ### Author Response · Authors · 2023-08-26
> **Responses to Reviewer GjRx - Part 2**
>
> Dear Reviewer GjRx,
>
> Thank you for your time and effort in reviewing this submission! We have tried our best to address the concerns and issues raised in the rebuttal. We also update the submitted papers and supplementary materials. Please let us know if you have any further questions, and we will be glad to clarify them. Thanks again for your valuable suggestions.
>
> Best, Authors.

---

### Official Review · Reviewer_APC2 · 2023-07-21
**Moderate dataset with some Limitations**

**Rating:** 5
**Confidence:** 5
**Correctness:** The evaluation seems correct to me.
**Clarity:** The paper is clear and well-written.

**Strengths:**

This paper has several notable strengths:
1. The writing is clear and well-structured. The authors state clearly what they aim to do, and this paper is easy to follow. The problem description is well done, as are description of experiments, which enhances the accessibility and accountability of the paper.
2. The article proposes a large dataset called GenImage, which contains one million fake images generated by eight different models. The authors have benchmarked several base models on two evaluation settings: Cross-Generator Image Classification and Degraded Image Classification. The use of this dataset will enable researchers to effectively develop and evaluate novel fake image detectors.


**Additional Feedback:**

N/A

**Documentation:**

The detail is sufficient for database construction.
However, a more detailed description of some important hyperparameters used for generation, such as guidance scale, sampling method and steps of diffusion, and seed should be added.

**Ethics:**

No issues. I agree with the statement written by the authors.

**Limitations:**

I think their contribution is limited:
1. Collecting such a fake image detection dataset is not a particularly difficult task, although it requires many GPU resources and will facilitate researchers' research. In my opinion, the contribution level of this dataset can only reach the level of a boardline accept.
2. The other limitation is that the datasets  focus on 8 types of models only, but in the future, there might be many other models coming out and this dataset approach doesn't seem scalable. The popular IF[1], CogView[2], for instance, is missing.
3. (Mentioned in Opportunities For Improvement) Based on the generation model, we can generate more fake images than just 1M (LAION has **2B** real images). Maybe more than releasing the dataset, it could make sense to release a modular code that users can use to generate more data using distributed inference with different generation models.
4. (Mentioned in Opportunities For Improvement) The author did not mention some important hyperparameters used for generation, such as guidance scale, sampling method and steps of diffusion, and seed.

[1] CogView: Mastering Text-to-Image Generation via Transformers

[2] https://github.com/deep-floyd/IF/tree/develop

**Opportunities For Improvement:**

I think there are some areas that can be improved:
1. The author did not mention some important hyperparameters used for Generation, such as guidance scale, sampling method and steps of diffusion, and random seed.
2. Maybe more than releasing the dataset, it could make sense to release a modular code that users can use to generate more data using distributed inference with different generation models.

**Relation To Prior Work:**

In my opinion, I think it's best to add the *related work section* in the main paper, although some articles have already been mentioned in the introduction.

**Summary And Contributions:**

This article proposes a large dataset GenImage which contains 1M fake images generated by 8 generation models. Authors also completes benchmarking some base models on two evaluation settings: Cross-Generator Image Classification and Degraded Image Classification. Researchers can effectively develope and evaluate novel fake image detectors with GenImage dataset. Overall, the paper is a valuable contribution to the field of fake image detection.

---

> ### Author Response · Authors · 2023-08-20
> **Responses to Reviewer APC2**
>
> Thanks to the reviewer for the valuable comments. We respond to the questions in the following.
> > **Q1.** The author did not mention some important hyperparameters used for Generation, such as guidance scale, sampling method and steps of diffusion, and random seed.
>
> **A1.** More details will be mentioned in the final version. The guidance scale of SD1.4, SD1.5, GLIDE, VQDM, and ADM are 7.5,7.5, 3.0, 1.0, and 10.0. The sampling method of SD1.4, SD1.5, and ADM are DDIM. The steps of diffusion of SD1.4, SD1.5, ADM, GLIDE, VQDM are 50,50, 1000,127,100. The random seed of SD1.4 and SD1.5 is 42. Some hyperparameters of these generators are not specified. We use the API of Midjourney and Wukong. Thus we cannot know the details of their hyperparameters.
>
> > **Q2.** Maybe more than releasing the dataset, it could make sense to release a modular code that users can use to generate more data using distributed inference with different generation models.
>
> **A2.** Thanks. We have uploaded the image generation code in the GitHub page.
>
> > **Q3.** Collecting such a fake image detection dataset is not a particularly difficult task, although it requires many GPU resources and will facilitate researchers' research. In my opinion, the contribution level of this dataset can only reach the level of a boardline accept.
> * **A3**. Thanks for the borderline accept. The contribution of GenImage not only includes collecting fake images but also exploring and analyzing the dataset. How to control and optimize the image combination is carefully considered in constructing GenImage. We propose a new benchmark and evaluate the existing detectors on it. We explore the influence of cross-generator and image degradation settings. We explore the influences of increasing the class and number of images. We demonstrate the generalization performance of the dataset.
>
> > **Q4.** The other limitation is that the datasets focus on 8 types of models only, but in the future, there might be many other models coming out and this dataset approach doesn't seem scalable. The popular IF[1], CogView[2], for instance, is missing.
> * **A4**. The models trained on eight generators are sufficient to generalize well on the other generative models. The experiments show that the detectors trained on GenImage can achieve high accuracy on representative models like IF [1], Cogview2 [2], NVAE [3], and StyleGAN [4]. The results will be added in the final version.
>
> |           | NVAE | CogView2 | StyleGAN | IF   |
> |:-:|:-:|:-:|:-:|:-:|
> | ResNet-50 | 93.4 | 97.5     | 97.9     | 90.2 |
>
> [1] https://github.com/deep-floyd/IF/tree/develop. 2023
>
> [2] "Cogview2: Faster and better text-to-image generation via hierarchical transformers." NIPS 2022.
>
> [3] "NVAE: A deep hierarchical variational autoencoder. NIPS 2020.
>
> [4] "A style-based generator architecture for generative adversarial networks." CVPR 2019.
>
> > **Q5.** In my opinion, I think it's best to add the related work section in the main paper, although some articles have already been mentioned in the introduction.
> *  **A5**. Thanks. We will add a related work section and move some content from the introduction section to this section. Due to the limitation of 5000 characters, the section can be reorganized as follows in short.
>
> 2.Related Work
>
> 2.1 AI-generated Image Detection Datasets.
>
> The rapid growth of generative neural networks has facilitated the generation of images, raising concerns over the propagation of misinformation due to the low-cost production of such images. Pioneering work on creating fake image detection datasets has made strides, such as CNNSpot [5], which solely employs ProGAN for training dataset generation and subsequently evaluates detector performance across a variety of GAN-based testing sets. A recent trend in this space involves the introduction of diffusion models as generators. For instance, based on a small-scale Cifar10 dataset [6], CIFAKE [7] generates fake images with Stable Diffusion V1.4. Nonetheless, these datasets often focus on GAN and they are data-limited.
>
> 2.2 AI-generated Image Detection Methods.
>
> Numerous approaches have emerged in the pursuit of spotting generated images. Spec [8] takes the frequency spectrum as input and synthesizes GAN artifacts in real images without the necessity of specific GAN models to generate fake images as training data. CNNSpot [5] uses ResNet-50 as a binary classifier, with specific pre- and post-processing and data augmentation. Most of the existing detectors are for GAN. The development of dedicated detectors tailored to the unique characteristics of mixed GAN and diffusion data is imperative.
>
> [5] "CNN-generated images are surprisingly easy to spot... for now." CVPR 2020.
>
> [6] "Learning multiple layers of features from tiny images." (2009): 7.
>
> [7] "CIFAKE: Image Classification and Explainable Identification of AI-Generated Synthetic Images." Arxiv 2023.
>
> [8] "Detecting and simulating artifacts in gan fake images." WIFS 2019.

---

> ### Author Response · Authors · 2023-08-26
> **Responses to Reviewer APC2 - Part 2**
>
> Dear Reviewer APC2,
>
> Thank you for your time and effort in reviewing this submission! We have tried our best to address the concerns and issues raised in the rebuttal. We also update the submitted papers and supplementary materials. Please let us know if you have any further questions, and we will be glad to clarify them. Thanks again for your valuable suggestions.
>
> Best, Authors.

---

### Official Review · Reviewer_27YU · 2023-07-24
**An intriguing dataset that could be scaled up and fleshed out**

**Rating:** 6
**Confidence:** 3

**Strengths:**

* Larger scale / more diverse than prior fake image detection datasets, which were either small-scale or focused just on faces.
* Includes recent diffusion models.
* The analysis of generalization (across generators and across image categories) is valuable.
* A timely contribution, as fake images are now ubiquitous on the Internet and there is large social interest in detecting them.

**Additional Feedback:**

Overall, I think this is a valuable dataset, but for it to meet the level of the conference, I feel like more should be done, along multiple dimensions:

* Only 8 generators were evaluated; there are by now many thousands of high quality generative models available online. The contribution would be much stronger if a much larger variety of generators were included.

* The benchmark could also be strengthened by including adversarial experiments and testing generalization as a function of the number of generators each detector is trained on.

* Most of these generators can make more kinds of images than just the 1000 ImageNet categories and faces. A more comprehensive dataset could be made that comes closer to testing _all_ the kinds of fake imagery these models can make. This could include not just class-conditional generation but also other kinds of generation, such generation from richer text descriptions and image-to-image generation.

If the dataset and paper can be scaled up along some or all of these dimensions, I think it will become much stronger.

**Clarity:**

The paper is mostly clear. I have just a few suggestions:

1. The paragraph on the details of cross-generator training and evaluaton is hard to follow (lines 160-182).
2. What does "noise residuals" mean on line 249?
3. Section 4.4 seems highly related to section 3.2; both are about cross-generator generalization. Perhaps they can be combined?
4. How is "Accuracy correlation" computed in Figure 3(b). What exactly is the difference between Figure 3(a) and (b)?

**Correctness:**

I did not find any significant errors. The dataset construction and evaluation appear to be sound.

**Documentation:**

The dataset itself is sufficiently documented. The GitHub page could be improved by providing a tutorial for usage, and by including code for training and benchmarking the detection methods (rather than just linking to their original repos).

**Ethics:**

I do not have concerns.

**Limitations:**

The limitations section is brief and could be expanded. I would appreciate discussion of the issue of adversaries, and limitations in terms of generalization to other generator architectures (beyond diffusion models and GANs).

**Opportunities For Improvement:**

* No analysis of robustness against adversaries. This is especially important for fake image detectors since any detector that is released would likely be attacked by adversaries trying to avoid being detected.
* Relatively little diversity in types of generative models; all are diffusion models except for BigGAN. This makes it hard to tell if the contribution is specific to diffusion models or is more general. I would like to see how well detectors trained on diffusion models generalize to other kinds of generators, such as autoregresive models, VAEs, or GANs that are more recent than BigGAN (e.g., StyleGAN).
* It would be interesting to analyze the effect of training a detector on more than one generative model. As the number of models you train against increases, does robustness or generality of the detector increase? Appendix A is a good start but does not evaluate if training on more models improves performance against _held out_ models.

**Relation To Prior Work:**

I feel the prior work is adequately covered. There is a longer history to fake image detection, but I don't think it's necessary to give a comprehensive survey here.

**Summary And Contributions:**

This paper introduces a new dataset for AI-generated image detection. The dataset follows the format of ImageNet: for each of 8 different generative models, 1.3 million images are sampled; ~1.3k per class for each of the 1k ImageNet classes. This dataset is used to benchmark several fake image detectors . The detectors are evaluated in terms of their ability to detect images made by the generators they are trained on, to detect images made by generators they were not trained on, to generalize to new image classes and categories (faces), and to detect fake images that have undergone JPEG degradation and blurring. The main advantage over prior fake image detection datasets is the scale and diversity of the data, and the incorporation of up-to-date diffusion models.

---

> ### Author Response · Authors · 2023-08-20
> **Responses to Reviewer 27YU**
>
> Thanks to the reviewer for the valuable comments. We respond to the questions in the following. The final version will be revised based on the following discussion.
> > **Q1.** No analysis of robustness against adversaries.
> * **A1.** Thanks. We use FGSM [1] and CW-L2 [2] for attacking methods. ResNet-50 is more robust for CW-L2 than FGSM.
>
> |          | FGSM | CW-L2 |
> |:-: |:-: |:-: |
> | ResNet50 | 88.0 | 99.9  |
>
> [1] "Explaining and harnessing adversarial examples." ArXiv 2014.
>
> [2] "Towards evaluating the robustness of neural networks." Ieee Symposium on Security and Privacy 2017.
>
> > **Q2.** I would like to see how well detectors trained on diffusion models generalize to other kinds of generators.
> * **A2.** ResNet-50 trained on all eight generators can generalize well on NVAE [3], CogView2 [4], StyleGAN [5], and IF[6].
>
> |           | NVAE | CogView2 | StyleGAN | IF   |
> |:-:|:-:|:-:|:-:|:-:|
> | ResNet50 | 93.4 | 97.5     | 97.9     | 90.2 |
>
> [3] "NVAE: A deep hierarchical variational autoencoder." NIPS 2020.
>
> [4] "Cogview2: Faster and better text-to-image generation via hierarchical transformers." NIPS 2022.
>
> [5] "A style-based generator architecture for generative adversarial networks." CVPR 2019.
>
> [6] https://github.com/deep-floyd/IF/tree/develop. 2023
>
> > **Q3.** As the number of models you train against increases, does robustness or generality of the detector increase?
> * **A3.** Thanks. The generality of the detector increase as the number of generators increases. One generator is SD V1.4. Four generators are SD V1.4, Midjourney, BigGAN, and ADM. Eight Generators are all the generators in GenImage.
>
> |                  | NVAE | CogView2 | StyleGAN | IF   |
> |:-:|:-:|:-:|:-:|:-:|
> | One Generator    | 64.2 | 79.0     | 65.7     | 62.4 |
> | Four Generators  | 70.4 | 95.6     | 68.1     | 82.8 |
> | Eight Generators | 93.4 | 97.5     | 97.9     | 90.2 |
>
> > **Q4.** The paragraph on the details of cross-generator training and evaluation is hard to follow.
> * **A4.** The cross-generator training and evaluation contains two settings. The first setting is training a model on SD V1.4, and evaluating the eight generators. The second setting is the same as the training and evaluation in Table 2. We average the eight average accuracies in Table 2 to obtain the results of ResNet-50 in Table 4.
>
> > **Q5.** What does "noise residuals" mean on line 249?
> * **A5.** The noise residuals R can be computed by the formula R = X- F(X), where X is the input image, and F(\dot) is the denoising filter DCNN [7].
>
> [7] "Beyond a gaussian denoiser: Residual learning of deep cnn for image denoising."TIP 2017.
>
> > **Q6.** Section 4.4 seems highly related to section 3.2; both are about cross-generator generalization. Perhaps they can be combined?
> * **A6.** In Section 3.2, we propose a benchmark that addresses the comparison of different methods. Section 4.4 mainly provides an intuitive and qualitative analysis of the correlation of generators. Since the emphasis is different, it seems separating them would be better.
>
> > **Q7.** How is "Accuracy correlation" computed in Figure 3(b). What exactly is the difference between Figure 3(a) and (b)?
> * **A7.** Figure 3(a) is the accuracy score map, which shows the accuracies with different training and testing approaches. In Figure 3(b), we compute Pearson correlation based on the accuracy score map. Pearson correlation coefficient is a measure of linear correlation between two sets of data.
>
> > **Q8.** The GitHub page could be improved by providing a tutorial for usage, and by including code for training and benchmarking the detection methods.
> * **A8.** The GitHub page has been updated. We include a tutorial for usage and the codes of detection methods.
>
> > **Q9.** The contribution would be much stronger if a much larger variety of generators were included.
> * **A9.** Thanks. The models trained on our eight generators are sufficient to generalize well on the other generative models. Please see **Q3** and **A3**.
>
> > **Q10.** This could include not just class-conditional generation but also other kinds of generation, such generation from richer text descriptions and image-to-image generation.
> * **A10.** The generalization performance has been evaluated on Face and Art Images and we conduct further experiments. Richer text descriptions experiment is based on prompts of CC12M [8]. For the image-to-image generation, we input the images and a template of "a painting of {class}" to the generator. For obtaining multi-object content images, the template is "photo of {class A}, {class B}, and {class C}". The models trained on eight generators are sufficient to generalize well on more kinds of images.
>
> |           | Richer Text Descriptions | Image-to-Image Generation | Multi-object Content |
> |:-:|:-:|:-:|:-:|
> | ResNet50 | 99.9                     | 99.3                      | 99.9                 |
>
> [8] "Conceptual 12m: Pushing web-scale image-text pre-training to recognize long-tail visual concepts." CVPR 2021.

---

> ### Author Response · Authors · 2023-08-26
> **Responses to Reviewer 27YU - Part 2**
>
> Dear Reviewer 27YU,
>
> Thank you for your time and effort in reviewing this submission! We have tried our best to address the concerns and issues raised in the rebuttal. We also update the submitted papers and supplementary materials. Please let us know if you have any further questions, and we will be glad to clarify them. Thanks again for your valuable suggestions.
>
> Best, Authors.

---

> > ### Comment · Reviewer_27YU · 2023-08-30
> > **thanks for the responses and improvements**
> >
> > Thanks for addressing my concerns and adding additional tests of robustness and generalization. I still feel like the paper is a bit light on content and could be expanded to include even more generators and more analysis along the lines of what I and other reviewers have suggested. However, I'll raise my score to borderine accept to reflect the improvements so far and because I think this dataset is a push in the right direction. I hope even more can be done and the authors continue to update the benchmark and dataset as time passes and new models get released.

---

### Author Response · Authors · 2023-08-22
**Global Response to All Reviewers**

We appreciate all reviewers for their valuable and constructive feedback. Separate responses to each reviewer are addressed as follows. Please note that we have made revisions to our paper and supplementary materials and updated them in the system. We mark changes in red in the paper and summarize them as follows. Ethics Reviewer proposes issues about the social impact, limitation, and ethical issues. We mention the consequences to use a detector trained on GenImage(lines 80-87 in supplementary material). We conduct experiments to explore the biased model problem for particular demographics (lines 68-78 in supplementary material). We upload "Datasheets for Datasets" documents(in supplementary material zip file). The ethical issues of the ImageNet dataset mentioned in the three related works have been summarized (lines 95-111 in supplementary material). Multiple reviews suggest further experiments regarding the generalization and robustness. The detector trained on GenImage is evaluated on other kinds of generators and more complex scenes and content (lines 36-43, 50-57 in supplementary material). We also analyze the robustness against adversaries (lines 31-34 in supplementary material). We show that the generality of the detector increase as the number of generators increases (line 45-48 in supplementary material). We clarify some details in the main paper (lines 214-217, 289-290, 333-336) and provide more content on the GitHub page. We have added a related work section (lines 68-104). We add the reason why the artifacts occur in GANs and the fact that the artifacts do not occur in Diffusion models in the main paper (lines 297-303 in the main paper). We are going to add an additional hidden test set and maintain a leaderboard. Thanks to all reviews' constructive suggestions, which have helped us improve this work.

---

### Decision · Program_Chairs · 2023-09-22

**Decision:**

Accept (Poster)

**Comment:**

This paper introduces a new dataset for AI-generated image detection. This dataset is used to benchmark several fake image detectors.
The dataset addresses an important and hot topic.
The proposal has some limitations, pointed out by the reviewers, but overall the evaluation is positive.